# MEMO: A Deep Network for Flexible Combination of Episodic Memories

**Andrea Banino**[1,2,†*]          **Adrià Puigdomènech Badia**[1,†]          **Raphael Köster**[1]

**Martin J. Chadwick**[1]     **Vinicius Zambaldi**[1]     **Demis Hassabis**[1]     **Caswell Barry**[3]

**Matthew Botvinick**[1]          **Dharshan Kumaran**[1,‡]          **Charles Blundell**[1,‡]

## ABSTRACT

Recent research developing neural network architectures with external memory have often used the benchmark bAbI question and answering dataset which provides a challenging number of tasks requiring reasoning. Here we employed a classic associative inference task from the memory-based reasoning neuroscience literature in order to more carefully probe the reasoning capacity of existing memory-augmented architectures. This task is thought to capture the essence of reasoning – the appreciation of distant relationships among elements distributed across multiple facts or memories. Surprisingly, we found that current architectures struggle to reason over long distance associations. Similar results were obtained on a more complex task involving finding the shortest path between nodes in a path. We therefore developed MEMO, an architecture endowed with the capacity to reason over longer distances. This was accomplished with the addition of two novel components. First, it introduces a separation between memories/facts stored in external memory and the items that comprise these facts in external memory. Second, it makes use of an adaptive retrieval mechanism, allowing a variable number of 'memory hops' before the answer is produced. MEMO is capable of solving our novel reasoning tasks, as well as match state of the art results in bAbI.

## 1 INTRODUCTION

During our every day life we need to make several judgments that require connecting facts which were not experienced together, but acquired across experiences at different points in time. For instance, imagine walking your daughter to a coding summer camp and encountering another little girl with a woman. You might conclude that the woman is the mother of the little girl. Few weeks later, you are at a coffee shop near your house and you see the same little girl, this time with a man. Based on these two separated episodes you might infer that there is a relationship between the woman and the man. This flexible recombination of single experiences in novel ways to infer unobserved relationships is called inferential reasoning and is supported by the hippocampus (Zeithamova et al., 2012).

Interestingly, it has been shown that the hippocampus is storing memories independently of each other through a process called pattern separation (Yassa & Stark, 2011; Marr et al., 1991). The reason hippocampal memories are kept separated is to minimize interference between experiences, which allows us to recall specific events in the form of 'episodic' memories (Eichenbaum & Cohen, 2004; Squire et al., 2004). Clearly, this separation is in conflict with the above mentioned role of the hippocampus in generalisation – i.e. how can separated memories be chained together? Interestingly, a recent line of research (Kumaran & McClelland, 2012; Banino et al., 2016; Schapiro et al., 2017; Koster et al., 2018) sheds lights on this tension by showing that the integration of separated experiences emerges at the point of retrieval through a recurrent mechanism. This allows

---

[*,1] DeepMind, London, UK. [2] CoMPLEX, University College London, London, UK. [3] Department of Cell and Developmental Biology, University College London, London, UK.
Correspondece should be sent to abanino@google.com, adriap@google.com, cblundell@google.com
†, ‡ These authors contributed equally.

multiple pattern separated codes to interact, and therefore support inference. In this paper we rely on these findings to investigate how we can take inspiration from neuroscience models to investigate and enhance inferential reasoning in neural networks.

Neural networks augmented with external memory, like the Differential Neural Computer (Graves et al., 2016, DNC), and end to end memory networks (Sukhbaatar et al., 2015, EMN) have shown remarkable abilities to tackle difficult computational and reasoning tasks. Also, more powerful attention mechanisms (Vaswani et al., 2017; Dehghani et al., 2018) or the use of context (Seo et al., 2016) have recently allowed traditional neural networks to tackle the same set of tasks. However, some of these tasks – e.g. bAbI (Weston et al., 2015) – present repetitions and commonalities between the train and the test set that neural networks can exploit to come up with degenerate solutions. To overcome this limitation we introduced a new task, called Paired Associative Inference (PAI - see below), which is derived from the neuroscientific literature (Bunsey & Eichenbaum, 1996; Banino et al., 2016). This task is meant to capture the essence of inferential reasoning – i.e. the appreciation of distant relationships among elements distributed across multiple facts or memories. PAI is fully procedurally generated and so it is designed to force neural networks to learn abstractions to solve previously unseen associations.

We then use the PAI task, followed by a task involving finding the shortest path and finally bAbi to investigate what kind of memory representations effectively support memory based reasoning. The EMN and other similar models (Sukhbaatar et al., 2015; Santoro et al., 2017; Pavez et al., 2018) have used fixed memory representations based on combining word embeddings with a positional encoding transformation. A similar approach has been recently implemented by current state of the art language model (Vaswani et al., 2017; Devlin et al., 2018). By contrast our approach, called MEMO, retains the full set of facts into memory, and then learns a linear projection paired with a powerful recurrent attention mechanism that enable greater flexibility in the use of these memories. MEMO is based on the same basic structure of the external memory presented in EMN (Sukhbaatar et al., 2015), but its new architectural components can potentially allow for flexible weighting of individual elements in memory and so supporting the form of the inferential reasoning outlined above.

Next, we tackle the problem of prohibitive computation time. In standard neural networks, the computation grows as a function of the size of the input, instead of the complexity of the problem being learnt. Sometimes the input is padded with a fixed number of extra values to provide greater computation (Graves et al., 2016), in other cases, input values are systematically dropped to reduce the amount of computation (e.g., frame dropping in reinforcement learning (Mnih et al., 2016)). Critically, these values are normally hand tuned by the experimenter; instead, here we are interested in adapting the amount of compute time to the complexity of the task. To do so we drawn inspiration from a model of human associative memory called REMERGE (Kumaran & McClelland, 2012). In this model, the content retrieved from memory is recirculated back as the new query, then the difference between the content retrieved at different time steps in the re-circulation process is used to calculate if the network has settled into a fixed point, and if so this process terminates.

To implement this principle in a neural network, we were inspired by techniques such as adaptive computation time (Graves, 2016). In our architecture, the network outputs an action (in the reinforcement learning sense) that indicates whether it wishes to continue computing and querying its memory, or whether it is able to answer the given task. We call this the halting policy as the network learns the termination criteria of a fixed point operator. Like ACT, the network outputs a probability of halting, but unlike ACT, the binary halting random variable is trained using REINFORCE (Williams, 1992). Thus we use reinforcement learning to adjust weights based upon the counterfactual problem: what would be the optimal number of steps of computation, given a particular number of steps was taken this time? The use of REINFORCE to perform variable amount of computation has been investigated already (e.g. Shen et al., 2017; Louizos et al., 2017) however our approach differs in that we added an extra term to the REINFORCE loss that, by exploiting the mathematical properties of binary random variables, naturally minimizes the expected number of computation steps. Thus we directly encourage our network to explicitly prefer representations and computation that minimize the amount of required computation.

To sum up, our contributions are:

1. A new task that stresses the essence of reasoning — i.e. the appreciation of distant relationships among elements distributed across multiple facts.

2. An in depth investigation of the memory representation that support inferential reasoning, and extensions to existing memory architectures that show promising results on these reasoning tasks.

3. A REINFORCE loss component that learn the optimal number of iterations required to learn to solve a task.

4. Significant empirical results on three tasks demonstrating the effectiveness of the above two contributions: paired associative inference, shortest path finding, and bAbI (Weston et al., 2015).

## 2 METHODS

### 2.1 RECAPITULATING END-TO-END MEMORY NETWORKS

We begin by describing End-to-End Memory Networks (Sukhbaatar et al., 2015, EMN), as a reminder of this architecture, to introduce notation and nomenclature, and also as a contrast to our work. We will focus on the multilayer, tied weight variant of EMN as this most closely resembles our architecture.

The set up used for the rest of the paper is as follows: given a set of knowledge inputs $\{x_i\}_{i=1}^I = \{\{x_{i1}, x_{i2}, ..., x_{iS}\}\}_{i=1}^I$, and a query or question $q = \{q_1, q_2, ..., q_S\} \in \mathbb{R}^S$, the network must predict the answer $a$. $I$ represents the length of the knowledge input sequence, and $S$ is the length of each input sentence; for instance, in bAbI (Weston et al., 2015), $I$ will be the number of stories and $S$ is the number of words in each sentence in a story. $x_{is}$ will be the word in position $s$ in the sentence, in the $i$th story and will be a $O$-dimensional one hot vector encoding one of $O$ possible input words.

EMN embeds each word and sums the resulting vectors:

$$k_i = \sum_s l_s \cdot W_k x_{is} \tag{1}$$

$$v_i = \sum_s W_v x_{is} \tag{2}$$

$$q_0 = W_q q \tag{3}$$

where $W_k, W_v \in \mathbb{R}^{d \times O}$, $W_q \in \mathbb{R}^{d \times S}$ are embedding matrices for the key, values and query, respectively. Also $l_s$ is a positional encoding column vector (as defined in (Sukhbaatar et al., 2015)), $\cdot$ represent an element wise multiplication and $O$ is the size of the vocabulary.

At each step $t$, EMN calculates the vector of weights over the memory elements $k_i$ and produces the output. Let $K$ be the $I \times d$ matrix formed by taking $k_i$ as its rows, and similarly $V$ formed by $v_i$ as rows, then:

$$w_t = \text{softmax}(K q_t) \tag{4}$$
$$q_{t+1} = w_t V + W_{qv} q_t \tag{5}$$
$$a_t = \text{softmax}(W_a q_{t+1}) \tag{6}$$

where $w_t \in \mathbb{R}^I$ are weights over the memory slots, $W_{qv}, W_a \in \mathbb{R}^{d \times d}$ is a linear mapping relating the query at the previous step to the current one, $q_{t+1}$ is the query to be used at the next step, and $a_t$ is the answer (usually only produced right at the end). EMN is trained via a cross entropy loss on $a_t$ at the final step.

### 2.2 MEMO

MEMO embeds the input differently. First, a common embedding $c_i$, of size $S \times d_c$, is derived for each input matrix $x_i \in \mathbb{R}^{S \times O}$:

$$c_i = x_i W_c \tag{7}$$

where $W_c \in \mathbb{R}^{O \times d_c}$. Then each of these embeddings is adapted to either be a key or a value. However, contrary to EMN, we do not use hand-coded positional embeddings, instead the words in

each sentence and their one-hot encoding in $x_i$, embedded as $c_i$, are combined and then this vector is passed trough a linear projection followed by an attention mechanism (explained in detail below). This allows $c_i$ to capture flexibly any part of the input sentence in $x_i$. MEMO uses multiple heads to attend to the memory following (Vaswani et al., 2017). Each head has a different view of the same common inputs $c_i$. Let $H$ denote the total number of heads, and $h$ index the particular head, then for each $h \in \{1, \ldots, H\}$ we have:

$$k_i^{(h)} = W_k^{(h)} \text{vec}(c_i) \tag{8}$$

$$v_i^{(h)} = W_v^{(h)} \text{vec}(c_i) \tag{9}$$

$$q_0^{(h)} = W_q^{(h)} q \tag{10}$$

where $W_k^{(h)}, W_v^{(h)} \in \mathbb{R}^{d \times Sd_c}$ and $W_q^{(h)} \in \mathbb{R}^{d \times S}$ are embedding matrices for the key, values and query respectively. $\text{vec}(c)$ means flattening the matrix $c$ into a vector with the same number of elements as the original matrix, and $\text{vec}^{-1}(v)$ is the reverse operation of a vector $v$ into a matrix such that $\text{vec}^{-1}(\text{vec}(c)) = c$. The result is three $d$-dimensional vectors $k_i^{(h)}$, $v_i^{(h)}$ and $q_0^{(h)}$. Keeping each item separated into memory allow us to learn how to weight each of these items when we perform a memory lookup. This contrasts with the hand-coded positional embeddings used in EMN (Sukhbaatar et al., 2015) and updated recently in Vaswani et al. (2017), and proved critical for enabling the flexible recombination of the stored items.

The attention mechanism used by MEMO also differs from that shown above for EMN. Firstly, it is adapted to use multi-head attention. Secondly, we use DropOut (Srivastava et al., 2014) and LayerNorm (Ba et al., 2016) to improve generalisation and learning dynamics. Let $K^{(h)} \in \mathbb{R}^{I \times d}$ denote the matrix formed by taking each $k_i^{(h)}$ as a row, and $V^{(h)}$ be the matrix formed by taking each $v_i^{(h)}$ as a row. In contrast, let $Q_t \in \mathbb{R}^{H \times d}$ be the matrix formed by taking each $q_t^{(h)}$ as the rows. The attention mechanism then becomes:

$$h_t^{(h)} = \frac{1}{\sqrt{d}} W_h K^{(h)} q_t^{(h)} \tag{11}$$

$$w_t^{(h)} = DropOut(\text{softmax}(h_t^{(h)})) \tag{12}$$

$$q_{t+1}^{(h)} = w_t^{(h)} V^{(h)} \tag{13}$$

$$Q_{t+1} = \text{LayerNorm}\left(\text{vec}^{-1}(W_q \text{vec}(Q_{t+1})) + Q_t\right) \tag{14}$$

$$a_t = \text{softmax}\left(W_a DropOut(\text{relu}(W_{qa} \text{vec}(Q_{t+1})))\right) \tag{15}$$

where $W_h \in \mathbb{R}^{I \times I}, W_q \in \mathbb{R}^{Hd \times Hd}$ are matrices for transforming the logits and queries respectively. $W_a \in \mathbb{R}^{O \times d_a}$ and $W_{qa} \in \mathbb{R}^{d_a \times Hd}$ are the matrices of the output MLP that produces the answer $a_t$. It is worth noting, that even though our attention mechanism uses some of the feature implemented in Vaswani et al. (2017) – i.e. normalization factor $sqrt(d)$ and multiheading, it differs from it because rather than doing self-attention it preserves the query separated from the keys and the values. This aspect is particularly important in terms of computational complexity, in that MEMO is linear with respect to the number of sentences of the input, whereas methods relying on self-attention (e.g. Dehghani et al., 2018) have quadratic complexity (see Appendix E).

## 2.3 THE HALTING POLICY

In the previous sections we described how MEMO can output a sequence of potential answers to an input query, here we describe how to learn the number of computational steps – hops – required to effectively answer $a$. To make this decision, we collect some information at every step and we use this to create an observation $s_t$. That observation is then processed by gated recurrent units (GRUs) (Chung et al., 2015) followed by a MLP which defines a binary policy $\pi(a|s_t, \theta)$ and approximate its value function $V(s_t, \theta)$. The input $s_t$ to this network is formed by the Bhattacharyya distance (Bhattacharyya, 1946) between the attention weights of the current time steps $W_t$ and the ones at the previous time step, $W_{t-1}$ (both $W_t$ and $W_{t-1}$ are taken after the softmax), and the number of steps taken so far as a one-hot vector $t$. The idea behind the way we build $s_t$ is that if the attention is focused on the same slot of memory for too many consecutive steps than there is no reason to keep

querying the memory because the information retrieved will be the same - i.e. the network has settled into a fixed point.

$$z_t = GRU_R(z_{t-1}, d(W_t, W_{t-1}), t) \tag{16}$$

$$v_t, \pi_t = MLP_R(z_t) \tag{17}$$

$$h_t = \sigma(\pi_t) \tag{18}$$

This network is trained using REINFORCE (Williams, 1992). More concretely, the parameters $\theta$ are adjusted using a $n$-step look ahead values, $\hat{R}_t = \sum_{i=0\ldots n-1} \gamma^i r_{t+i} + \gamma^n V(s_{t+n}, \theta)$, where $\gamma$ is the discount factor. The objective function of this network is to minimize: $\mathcal{L}_{\text{Hop-Net}} = \mathcal{L}_\pi + \alpha \mathcal{L}_V + \beta \mathcal{L}_{\text{Hop}}$ where: $\mathcal{L}_\pi = -\mathbb{E}_{s_t \sim \pi} \left[ \hat{R}_t \right]$, $\mathcal{L}_V = \mathbb{E}_{s_t \sim \pi} \left[ \left( \hat{R}_t - V(s_t, \theta) \right)^2 \right]$ and $\mathcal{L}_{\text{Hop}} = -\mathbb{E}_{s_t \sim \pi} \left[ \pi(\cdot | s_t, \theta) \right]$. Interestingly, $\mathcal{L}_{\text{Hop}}$ is a term that directly follows from the fact that $\pi$ is a binary policy. Specifically, the expectation of a binary random variable is its probability and the expectation of their sum is the sum of the expectation. Consequently, the new term that we introduce in the loss, $\mathcal{L}_{\text{Hop}}$ allows us to directly minimize the expected number of hops. This term, similar in motivation to (Louizos et al., 2017) (although differs mathematically), directly encourages our network to explicitly prefer representations and computation that minimise the amount of required computation. It is also worth noting that an argument against using REINFORCE when training discrete random variables is that the variance can be prohibitively high (Sutton & Barto, 2018). Interestingly, in the case of a binary halting random variable, the variance is just $p(1-p)$ where $p$ is the probability of halting and the variance is bounded by $1/4$ which we find is not too large for learning to proceed successfully in practice.

Finally, the reward structure is defined by the answer $a$:

$$r_t = \begin{cases} 1, & \text{if } \hat{a} = a \\ 0, & \text{otherwise} \end{cases}$$

where $a$ is the target answer associate with the input and $\hat{a}$ is the prediction from the network. The final layer of $MLP_R$ was initialized with $bias_{init}$, in order to increase the chances that $\pi$ produces a probability of 1 (i.e. do one more hop). Finally, we set a maximum number of hops, $N$, that the network could take. If $N$ was reached, the network stopped performing additional hops. Critically, there was no gradient sharing between the hop network and the main MEMO network explained above. All model hyperparameters are reported in appendix D.

## 3 RELATED WORK

### 3.1 MEMORY-AUGMENTED NEURAL NETWORKS

In recent years there has been increasing interest in the development of memory-augmented networks, due to their potential for solving abstract and relational reasoning tasks. Alongside the EMN (described in detail above), another influential early model deploying memory-augmentation was the Differential Neural Computer (Graves et al., 2014; 2016, DNC). The DNC operates sequentially on the inputs, and learns to read and write to a memory store. The model proved capable of solving a range of algorithmic problems, but was difficult to scale to higher-dimensional problem domains. A more recent extension incorporated sparsity into the DNC(Rae et al., 2016), allowing the model to perform well at larger-scale tasks, including the bAbI task suite (Weston et al., 2015). Since these initial models were published, a number of alternative memory-augmented architectures have been developed (Kumar et al., 2016; Henaff et al., 2016; Pavez et al., 2018). The Dynamic Memory Network (Kumar et al., 2016) shares some similarities with EMNs, but operates on sequential inputs rather than parallel. The Recurrent Entity Network (Henaff et al., 2016) has similarities with the DNC, but uses a parallel architecture, enabling simultaneous updates across several memory locations. The Working Memory Network (Pavez et al., 2018) is closely based on EMNs, but additionally incorporates a working memory buffer, and a RelationNet (Santoro et al., 2017). These components thereby enable relational reasoning over the retrieved contents of memory. Each of these new models has proven to perform well at various reference tasks, including the bAbI task suite (Weston et al., 2015).

### 3.2 Adaptive computation time

In general, the time required to solve a problem is expected to increase with the complexity of the task. However, most of the machine learning algorithms do not adapt their computational budget based on the complexity of the task. One approach to this problem is represented by Adaptive Computation Time (ACT) (Graves, 2016). ACT is a mechanism for learning a scalar halting probability, called the "ponder time", to dynamically modulate the number of computational steps needed for each input. An alternative approach is represented by Adaptive Early Exit Networks (Bolukbasi et al., 2017), which give the network the ability to exit prematurely - i.e. not computing the whole hierarchy of layers - if no more computation is needed. Another approach to conditional computation is the use of REINFORCE (Williams, 1992) to learn a discrete latent variables which dynamically adjust the number of computation steps. This has been applied to recurrent neural networks where each layer decides whether or not to activate the next one (Chung et al., 2016). REINFORCE has also been used to learn how many steps to "jump" in sequence, and so reducing the total number processed inputs (Yu et al., 2017). This jump technique has also been applied to recurrent neural network without the need of REINFORCE (Campos Camunez et al., 2018). A similar idea has also been applied to neural network augmented with external memeory (Shen et al., 2017), but it differs from ours both in the REINFORCE loss and in the fact that our method introduce the idea of using the distance between attention weights at different time steps (hop) as a proxy to check if more information can be retrieved from memory - i.e. another hop is needed – or whether the network has settled and it has enough information to correctly answer the query.

### 3.3 Graph Neural Networks

Graph Neural Networks (GNNs) (Scarselli et al., 2008; Gori et al., 2005) consist of an iterative message passing process which propagates node and edge embeddings throughout a graph. Neural networks are then used to aggregate or learn functions over graph components to perform supervised, semi-supervised, representation and reinforcement learning tasks (Kipf & Welling, 2016; Gilmer et al., 2017; Hamilton et al., 2017; Zambaldi et al., 2018). The message passing process implements similar computation to attention mechanisms (Veličković et al., 2017; Battaglia et al., 2018) and as a limit case, self-attention can be viewed as a fully-connected GNN. Our work differs from GNNs in two fundamental ways: even though GNNs may exhibit a recurrent component (Li et al., 2015; 2018), its implementation is based in unrolling the recurrence for a fixed number of steps and use backpropagation through time in the learning process, whereas our method performs an adaptive computation to modulate the number of message passing steps; our model does not require message passing between memories–input queries attend directly to the memory slots.

## 4 Paired associative inference task

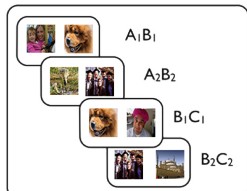 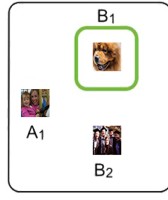 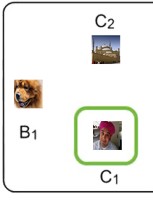 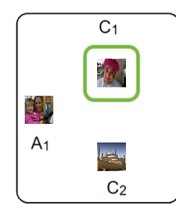

Figure 1: Paired associative inference. The panel on the left illustrates a memory store filled with random pairs of images. The panels to the right illustrate (from left to right) two 'direct' queries (AB and BC) where no inference is require, and an 'indirect' query (AC) where inference is required

One contribution of this paper is to introduce a task, derived from neuroscience, to carefully probe the reasoning capacity of neural networks. This task is thought to capture the essence of reasoning – the appreciation of distant relationships among elements distributed across multiple facts or memories. This process is formalized in a prototypical task widely used to study the role of the hippocampus in generalization – the paired associative inference (PAI) task (Bunsey & Eichenbaum, 1996; Banino

et al., 2016, see Fig. 1). Here, two images are randomly associated together. For example, analogous to seeing a little girl with a woman as in the example in the main text, in the PAI task the agent (human participant or a artificial neural network) would be presented with an image $A$, e.g. a woman, and image $B$, e.g. a girl, side by side. Later, in a separate event, the agent would be exposed to a second pair, the image $B$ again, but this time paired with a new image $C$, e.g. the other person. This is analogous to seeing the little girl a second time with a different person. During test time two types of query can be asked: direct and indirect queries. Direct queries are a test of episodic memory as the answer relies on retrieving an episode that was experienced. In contrast, indirect queries require inference across multiple episodes (see appendix A.1 for further details). Here the network is presented with CUE, image $A$, and two possible choices: image C, the MATCH, that was originally paired with B; or another image $C'$, the LURE, which was paired with $B'$ – i.e. forming a different triplet $A' - B' - C'$. The right answer, C, can only be produced by appreciating that $A$ and $C$ are linked because they both were paired with $B$. This is analogous to the insight that the two people walking the same little girl are likely to have some form of association. For the specific details on how the batch was created please refer to appendix A.1.

## 5   RESULTS

### 5.1   BASELINES

We compared MEMO with two other memory-augmented architectures: End to End Memory Networks (EMN) (Sukhbaatar et al., 2015) and DNC (Graves et al., 2016). We also compared MEMO with the Universal Transformer (UT) (Dehghani et al., 2018), the current state of the art model in the bAbI task suite (Weston et al., 2015). See the appendix for details about the baselines.

### 5.2   PAIRED ASSOCIATIVE INFERENCE

Table 1 reports the summary of results of our model (MEMO) and the other baselines on the hardest inference query for each of the PAI tasks (full results are reported in the Appendix A.2). On the smaller set - i.e. A-B-C - MEMO was able to achieve the highest accuracy together with DNC, whereas EMN, even with 4 or 10 hops, wasn't able to achieve the same level of accuracy, also UT was not able to accurately solve this inference test. For longer sequences -i.e. length 4 and 5 - MEMO was the only architecture which successfully answered the most complex inference queries.

To investigate how these results were achieved, we run further analysis on the length 3 PAI task. Interestingly, to solve this task DNC required 10 pondering steps to get the same accuracy as MEMO, which instead converged to 3 hops (see Fig. 3b in Appendix A.3). To analyze how MEMO approached this task we then analyzed the attention weights of an inference query, where the goal was to associate a CUE with the MATCH and avoid the interference of the LURE (see Appendix A.1 for task details). For clarity we report here the original sequence $A - B - C$, respectively composed by the following class IDs: $611 - 191 - 840$ (however this sequence was not directly experienced together by the network, as the two associations $A - B$ and $B - C$ where stored in slot 10 and 25, respectively). As depicted in Figure 2, in the first hop MEMO retrieved the memory in slot 10, which contained the CUE, ID 611, and the associated item, ID 191, which form an $A - B$ association. Then in the following hop this slot was partially active, but most of the mass was placed on slot 16, which contained the memory association $B - C$; that is, ID 191 and ID 840, the MATCH. Interestingly, slot 13 which was associated with the LURE, ID 943, got a bit of mass associated with it. So in this second hop MEMO assigned appropriate probability masses to all the slots needed to support a correct inference decision, which was then confirmed in the last hop. This sequence of memories activation is reminiscent of the one predicted by computational model of the hippocampus (Kumaran & McClelland, 2012; Schapiro et al., 2017) and observed in neural data (Koster et al., 2018). Moreover, another instance of MEMO which used 7 hops, to solve the task with the same level of accuracy, presented a very different pattern of memories activation (see Fig. 4 Appendix A.3.2). This is an indication of the fact that the the algorithm used to solve the inference problem depends on how many hops the network takes. This aspect could also be related to knowledge distillation in neural networks (Hinton et al., 2015; Frankle & Carbin, 2018), whereby many hops are used to initially solved the task (i.e. over-parametrization) and then these are automatically reduced to use less computation (see Fig. 3b in Appenix A.3).

We also ran a set of ablation experiments on MEMO (see Table 7 in Appendix A.3.1). This analysis confirmed that it is the combination of the specific memory representations (i.e. facts kept separated) and the recurrent attention mechanism that supports successful inference – i.e. employing these two components individually was not enough. Interestingly, this conclusion was valid only for inference queries, not for direct queries (see Fig. 3c,d in Appendix A.3). Indeed, by definition, direct queries are a pure test of episodic memory and so can be solved with a single memory look-up. Finally, we also compared our adaptive computation mechanism with ACT (Graves, 2016) and we found that, for this task, our method was more data efficient (see Fig. 5 in Appendix A.3.3).

Table 1: Inference queries

| Length | EMN | DNC | UT | MEMO |
|---|---|---|---|---|
| 3 items (set: A-B-C - accuracy on A-C) | 61.01 | 96.85 | 85.60 | 98.26(0.67) |
| 4 items (set: A-B-C-D - accuracy on A-D) | 48.66 | 51.56 | 44.16 | 97.22(0.13) |
| 5 items (set: A-B-C-D-E - accuracy on A-E) | 45.13 | 62.61 | 47.93 | 84.54(5.72) |

Test results for the best 5 hyper-parameters (chosen by validation loss) for MEMO. For EMN and DNC and UT results on the best run (chosen by validation loss)

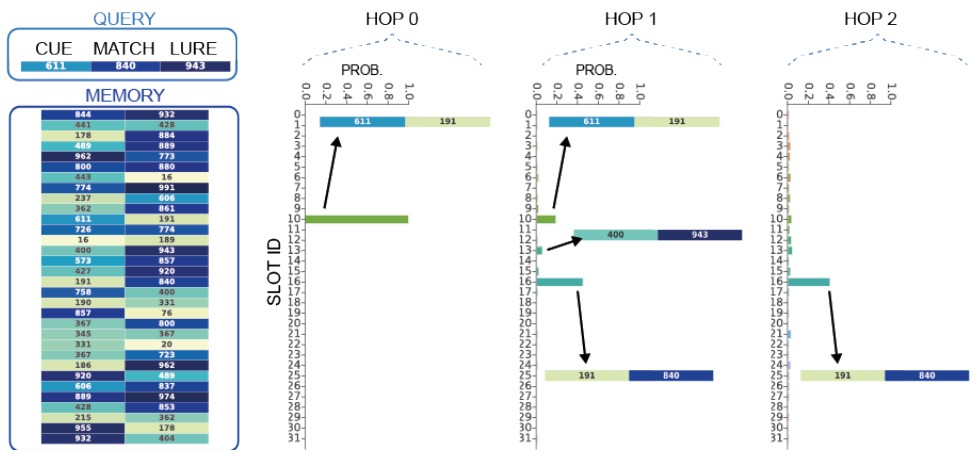

Figure 2: Weights analysis of an inference query in the length 3 PAI task. An example of memory content and related inference query is reported in the first column on the left. For clarity we report image class ID. Cue and Match are images from the same sequence e.g. $A_{10} - C_{10}$, where 10 is the slot ID. The lure is an images presented in the same memory store, but associated with a different sequence, e.g. $C_{13}$. The 3 most right columns report the weights associated with the 3 hops used by the network, for each probability mass we report the associated retrieved slot.

## 5.3 SHORTEST PATH ON RANDOMLY GENERATED GRAPHS

We then turn to a set of synthetic reasoning experiments on randomly generated graphs (see Appendix B.1 for details). Table 2 shows the accuracy of the models on the task related to finding the shortest path between two nodes. On a small graph with 10 nodes, with a path length of 2 and 2 outgoing edges per node, DNC, the Universal Transformer, and MEMO had perfect accuracy in predicting the intermediate shortest path node. However, on more complicated graphs (20 nodes, 3 separated outgoing edges), with a path length of 3, MEMO outperformed EMN in predicting the first node of the path (by more than 50%), and, similarly to DNC, almost completely solved the task. Additionally, MEMO outperformed DNC in more complicated graphs with a high degree of connectivity (5 out-degree), being better by more than 20% at predicting both nodes in the shortest path. This showed the great scalability that MEMO holds; the model was able to iterate and considered more paths as the number of hops increases. Finally, Universal Transformer had a different performance in predicting the first node versus the second one of the shortest path. In the latter case, the results showed that

UT achieved a slightly lower performance than MEMO, showing its great capability of computing operations that require direct reasoning.

Table 2: Undirected graph - shortest path

| Graph Structure | | | Prediction of First Node | | | | Prediction of Second Node | | | |
|---|---|---|---|---|---|---|---|---|---|---|
| Nodes | Out-degree | Path length | EMN | UT | DNC | MEMO | EMN | UT | DNC | MEMO |
| 10 | 2 | 2 | 94.99 | 100.00 | 100.00 | 100.00(0.00) | n/a | n/a | n/a | n/a |
| 20 | 3 | 3 | 31.99 | 39.00 | 97.00 | 94.40(0.02) | 66.00 | 84.80 | 98.00 | 93.00(0.03) |
| 20 | 5 | 3 | 23.99 | 28.00 | 30.00 | 69.20(0.07) | 43.00 | 61.51 | 40.99 | 68.80(0.09) |

Test results for the best 5 hyper-parameters (chosen by training loss) for MEMO, mean and corresponding standard deviation are reported. For EMN, UT, and DNC we report results from the best run.

### 5.4 QUESTION ANSWERING ON THE BABI TASKS

Finally, we turn our attention to the bAbI question answering dataset (Weston et al., 2015), which consists of 20 different tasks. In particular we trained our model on the joint 10k training set (specifics of training are reported in the appendix C.1).

Table 3 reports the averaged accuracy of our model (MEMO) and the other baselines on bAbI (the accuracy for each single task averaged across the best set of hyper-parameters is reported in Table 10 in the Appendix C.3). In the 10k training regime MEMO was able to solve all tasks, thereby matching the number of tasks solved by (Seo et al., 2016; Dehghani et al., 2018), but with a lower error (for single task results refer to Appendix C.3).

We also ran an extensive set of ablation experiments to understand the contribution of each architectural component to the final performance (see Appendix C.2 for details). As observed previously it was the combination of memory representations and the powerful recurrent attention that was critical to achieve state of the art performance on bAbI. Finally the use of layernorm (Ba et al., 2016) in recurrent attention mechanism was critical to achieve a more stable training regime and so better performance.

Table 3: bAbI - joint training

| Length | Memory Networks | DNC | Universal Transformer | Memo |
|---|---|---|---|---|
| 10k | 4.2 (17/20) | 3.8 (18/20) | 0.29 (20/20) | 0.21 (20/20) |

Test results for the best run (chosen by validation loss) on the bAbI task. DNC results from (Graves et al., 2016), Universal Transformer results from (Dehghani et al., 2018). In parenthesis we report the number of tasks solved, with 20 being the maximum.

## 6 CONCLUSIONS

In this paper we conducted an in-depth investigation of the memory representations that support inferential reasoning and we introduce MEMO, an extension to existing memory architectures, that shows promising results on these reasoning tasks. MEMO showed state-of-the-art results in a new proposed task, the paired associative inference, which had been used in the neuroscience literature to explicitly test the ability to perform inferential reasoning. On both this task, and a challenging graph traversal task, MEMO was the only architecture to solve long sequences. Also, MEMO was able to solve the 20 tasks of the bAbI dataset, thereby matching the performance of the current state-of-the-art results. Our analysis also supported the hypothesis that these results are achieved by the flexible weighting of individual elements in memory allowed by combining together the separated storage of single facts in memory with a powerful recurrent attention mechanism.

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

ACKNOWLEDGMENTS

The authors would like to thank Adam Santoro, and many other colleagues at DeepMind for useful discussions and feedback on this work.

# A  APPENDIX

## A.1  PAIRED ASSOCIATIVE INFERENCE TASK

(Please refer to Figure 1 in the main text)

To make this task challenging for a neural network we started from the ImageNet dataset (Deng et al., 2009). We created three sets, training, validation and test which used the images from the respective three sets of ImageNet to avoid any overlapping. All images were embedded using a pre-trained ResNet (He et al., 2016). We generated 3 distinct datasets with sequences of length three (i.e. $A - B - C$), four (i.e. $A - B - C - D$) and five (i.e. $A - B - C - D - E$) items. Each dataset contains $1e6$ training images, $1e5$ evaluation images and $2e5$ testing images. Each sequence was randomly generate with no repetition in each single dataset.

To explain how the batch was built let's refer to sequences of length, $S$, being equal to 3. Each batch entry is composed by a memory, a query and a target. In order to create a single entry in the batch we selected $N$ sequences from the pool, with $N = 16$.

First, we created the memory content with all the possible pair wise association between the items in the sequence, e.g. $A_1B_1$ and $B_1C_1$, $A_2B_2$ and $B_2C_2$, ..., $A_NB_N$ and $B_NC_N$. For $S = 3$, this resulted in a memory with 32 rows.

Then we generated all the possible queries. Each query consist of 3 images: the cue, the match and the lure. The cue is an image from the sequence (e.g. $A_1$), as is the match (e.g. $C_1$). The lure is an image from the same memory set but from a different sequence (e.g. $C_7$). There are two types of queries - 'direct' and 'indirect'. In 'direct' queries the cue and the match can be found in the same memory slot, so no inference is require. For example, the sequence $A_1$ - $B_1$ - $C_1$ produces the pairs $A_1$ - $B_1$ and $B_1$ - $C_1$ which are stored different slots in memory. An example of a direct test trail would be $A_1$ (cue) - $B_1$ (match) - $B_3$ (lure). Therefore, 'direct' queries are a test of episodic memory as the answer relies on retrieving an episode that was experienced. In contrast, 'indirect' queries require inference across multiple episodes. For the previous example sequence, the inference trail would be $A_1$ (cue) - $C_1$ (match) - $C_3$ (lure). The queries are presented to the network as a concatenation of three image embedding vectors (the cue, the match and the lure). The cue is always in the first position in the concatenation, but to avoid any degenerate solution, the position of the match and lure are randomized. It is worth noting that the lure image always has the same position in the sequence (e.g. if the match image is a C the lure is also a C) but it is randomly drawn from a different sequence that is also present in the current memory. This way the task can only be solved by appreciating the correct connection between the images, and this need to be done by avoiding the interference coming for other items in memory. For each entry in the batch we generated all possible queries that the current memory store could support and then one was selected at random. Also the batch was balanced, i.e. half of the elements were direct queries and the other half was indirect. The targets that the network needed to predict are the class of the matches.

It is worth mentioning that longer sequences provide more 'direct' queries, but also multiple 'indirect' queries that require different levels of inference, e.g. the sequence $A_n$ - $B_n$ - $C_n$ - $D_n$ - $E_n$ produces the 'indirect' trial $A_1$ (cue) - $C_1$ (target) - $C_3$ (lure) with 'distance' 1 (one pair apart) and $A_1$ (cue) - $E_1$ (target) - $E_3$ (lure) with 'distance' 4 (4 pairs apart). The latter trial required more inference steps and requires to appreciate the overlapping images of the entire sequence.

Finally we use the inputs as follows:

- For EMN and MEMO, memory and query are used as their naturally corresponding inputs in their architecture.

- In the case of DNC (section G), we embed stories and query in the same way it is done for MEMO. Memory and query are presented in sequence to the model (in that order), followed by blank inputs as pondering steps to provide a final prediction.

- For UT, we embed stories and query in the same way it is done for MEMO. Then we use the encoder of UT with architecture described in Section H. We use its output as the output of the model.

## A.2 PAI - QUERIES WISE RESULTS

The result repoted below are from the evaluation set at the end of training. Each evaluation set contains 600 items.

Table 4: Paired Associative - length 3: A-B-C

| Trial Type | EMN | DNC | UT | MEMO |
|---|---|---|---|---|
| A-B | 98.19 | 98.58 | 97.43 | 99.82(0.30) |
| B-C | 97.93 | 99.34 | 98.28 | 99.76(0.38) |
| A-C | 61.01 | 96.85 | 85.60 | 98.26(0.67) |

Table 5: Paired Associative - length 4: A-B-C-D

| Trial Type | EMN | DNC | UT | MEMO |
|---|---|---|---|---|
| A-B | 96.31 | 94.26 | 99.32 | 99.57(0.20) |
| B-C | 97.57 | 84.94 | 88.31 | 99.33(0.13) |
| C-D | 96.59 | 95.68 | 93.37 | 99.58(0.13) |
| A-C | 48.71 | 49.38 | 54.87 | 98.93(0.15) |
| B-D | 47.42 | 49.89 | 51.92 | 99.14(0.19) |
| A-D | 48.63 | 58.63 | 44.16 | 97.22(0.13) |

Table 6: Paired Associative - length 4: A-B-C-D-E

| Trial Type | EMN | DNC | UT | MEMO |
|---|---|---|---|---|
| A-B | 95.68 | 98.88 | 96.94 | 99.20(0.43) |
| B-C | 95.82 | 94.60 | 92.63 | 98.93(0.17) |
| C-D | 95.43 | 95.20 | 89.99 | 97.27(0.21) |
| D-E | 95.16 | 95.98 | 97.27 | 95.06(0.12) |
| A-C | 48.68 | 48.66 | 41.85 | 87.33(0.12) |
| B-D | 45.75 | 46.87 | 39.62 | 86.65(1.27) |
| C-E | 49.46 | 49.51 | 35.87 | 87.08(0.92) |
| A-D | 52.08 | 50.32 | 52.38 | 86.12(0.57) |
| B-E | 46.69 | 52.27 | 43.27 | 86.37(0.77) |
| A-E | 48.31 | 48.79 | 47.93 | 84.54(5.72) |

For EMN and DNC the results shown are from the hyper-parameter with the lower loss on the validation set. For MEMO the results are the average and relative standard deviation (reported in parenthesis), obtained by averaging the 5 hyper-parameters with the lower loss on the validation set.

### A.3 PAIRED ASSOCIATIVE INFERENCE FURTHER ANALYSIS ON LENGTH 3 EXPERIMENTS.

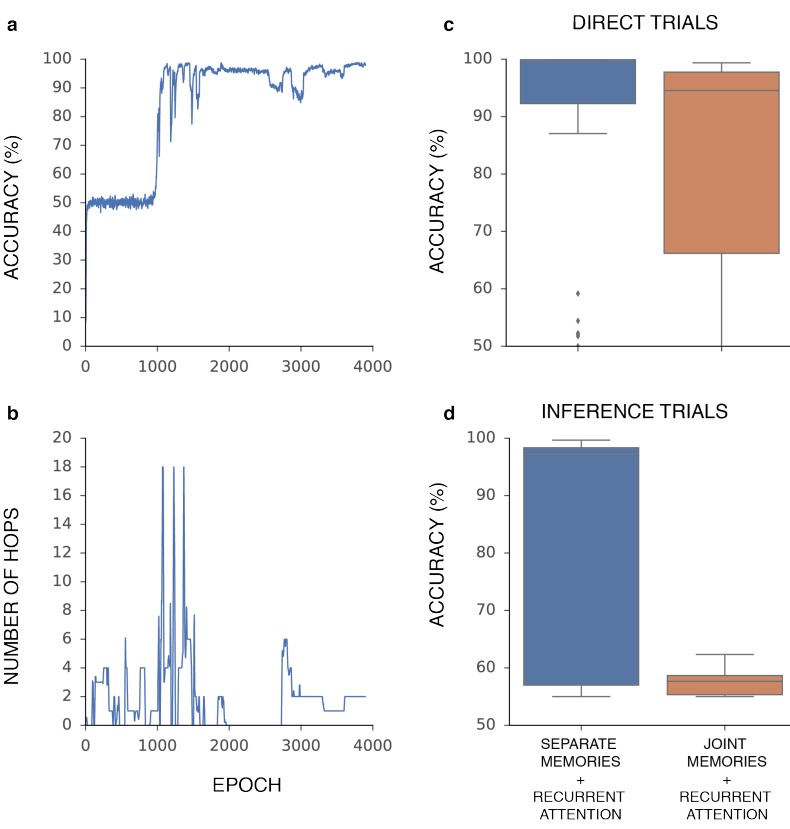

Figure 3: Analysis of length 3 PAI task. a. Evaluation accuracy on the inference trial A-C; b. Number of hops taken during training; c. Distribution of evaluation accuracy obtained by averaging direct queries (A-B and B-C). This was obtained over 100 different hyper-parameters and seeds; d. same as c, but on the inference queries (A-C)

### A.3.1 ABLATIONS

Table 7: PAI - Ablations - sequence of length 3: A-B-C

| MEMO Network Architecture | | | A-C inference trial |
| --- | --- | --- | --- |
| Positional encoding as in (Vaswani et al., 2017) | Memories kept separated | Recurrent attention w/ Layernorm | Accuracy |
| ✓ | ✗ | ✗ | 57.59(10.11) |
| ✓ | ✗ | ✓ | 52.79(3.12) |
| ✗ | ✓ | ✗ | 73.26(15.86) |
| ✗ | ✓ | ✓ | 97.59(1.85) |

Results for the best run (chosen by validation set) on the PAI task. ✗= not present; ✓= present

### A.3.2 ATTENTION WEIGHTS ANALYSIS

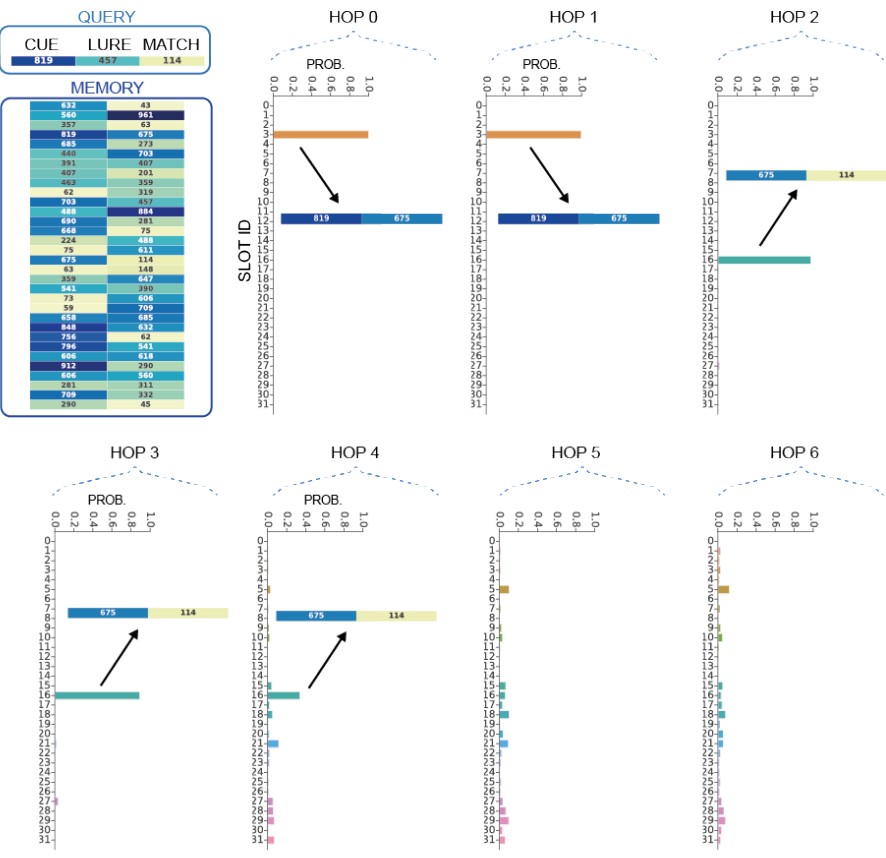

Figure 4: Attention weights analysis of length 3 PAI task, in the case where the network converged to 7 hops. In this case the network uses the first two hops to retrieve the slot where the cue is present and the the hops number 3, 4 and 5 to retrieve the slot with the match. The weights are sharp and they focus only on 1 single slot.

### A.3.3 ADAPTIVE COMPUTATION

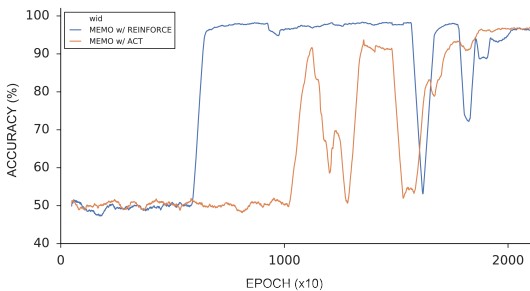

Figure 5: Comparison between MEMO + REINFORCE and MEMO + ACT on length 3 PAI task. MEMO wit REINFORCE shows more data efficiency that the one where the adaptive computation is done with ACT.

## B SHORTEST PATH TASK

### B.1 TRAINING DETAILS

**Graph generation**    In the shortest path experiments, we generate the graph in the same fashion as Graves et al. (2016): the graphs used to train the networks are generated by uniformly sampling a set of two-dimensional points from a unit square, each point corresponding to a node in the graph. For each node, the K nearest neighbours in the square are used as the $K$ outbound connections, with $K$ independently sampled from a uniform range for each node.

**Graph representation**    We represent our task in three parts: a graph description, a query, and the target. The graph description is presented a sequence of tuples of integers that represent a connection between nodes, holding a token for the source node, and another for the destination node. The query is also represented as a tuple of integers, although, in that case, source and destination are simply the beginning and end of the path to find. The target is the sequence of node IDS that constitute the path between source and destination of the query.

When training, we sample a mini-batch of $64$ graphs, with associated queries and target paths. Following our description above, queries are represented as a matrix of size $64 \times 2$, targets are of size $64 \times (L - 1)$, and graph descriptions are of size $64 \times M \times 2$, where $L$ is the length of the shortest path, and $M$ is the number of maximum nodes we allow one graph description to have. In our experiments, we fix the upper bound $M$ to be the maximum number of nodes that we have multiplied by the out-degree of the nodes in the graph.

All networks were trained for $2e4$ epochs, each one formed by $100$ batch updates.

- For EMN and MEMO, we set the graph description to be the contents of their memory, and we use the query as input. In order to answer the sequence of nodes that is used as target, we keep the keys $k_i^{(h)}$ and values $v_i^{(h)}$ fixed, and we proceed to use our algorithm as described for each answer, with independent numbers of hops for each one. The model then predict the answers for nodes sequentially: the first node is predicted before the second. However, one important difference between MEMO and EMN is that for EMN we use the ground truth answer of the first node as the query for the second node, whereas for MEMO we used the answer predicted by the model for the first node as the query for the second node. This was done to enhance the performance of EMN while testing the real capabilities of MEMO to sequentially reasoning over multiple steps problems. The weights that are used for each answer are not shared.

- For the Universal Transformer, we also embed the query and graph description as done for EMN and MEMO. After that, we concatenate the embeddings of query and graph description and use the encoder of the UT architecture (with specific description in Section H). We use

its output as answer. After providing an answer, that answer is provided as initial query for the following round of hops. The weights that are used for each answer are not shared.

- For DNC, we also embed the query and graph description as done for EMN and MEMO. Since it is naturally a sequential model, the information is presented differently: the tuples of the graph description are presented first, and after that the query tuple is presented. After that, the pondering steps are used to be able to output the sequence of nodes that constitute the proposed shortest path.

The output of the models is trained using Adam using a cross-entropy loss against all the sampled target sequences. Training is done for a fixed number of steps, detailed in Appendix Section D.2.

For evaluation, we sample a batch of 600 graph descriptions, queries, and targets. We evaluate the mean accuracy over all the nodes of the target path. We report average values and standard deviation over the best 5 hyper parameters we used.

It is worth noting that given this training regime:

- DNC and UT have a 'global view' on the problem in order to provide an answer for the second node. This means that, to answer the second node in the path, they can still reason and work backwards from the end node, and while still having information about the initial node in the path. This makes it intuitive for them to achieve better performance in the second node, as it is closest to the end node of the path, so less reasoning is needed to achieve good performance.

- On the contrary, MEMO has a 'local view' on the problem, the answer to the second node depends on the answer about the first node. Therefore, it cannot do better than chance if the answer to the first node is not correct.

## B.2 RESULTS

To better compare the performance of MEMO versus EMN we also run another experiment where we tested the models in two conditions:

- The ground truth answer of the first node was used as the query for the second node.

- The answer predicted by the model for the first node was used as the query for the second node.

The results are summarized in Table 8. In the case of 20 Nodes with 5 outbound edges, we can see that if we give MEMO the ground truth for node 1 as query for node 2 the performance increases from the one related to the prediction of the first node (85.38%(0.05) vs. 69.20%(0.07)). Interestingly, if we use for EMN the same training regime used for MEMO - i.e. the prediction is used to query the second node - then EMN perform almost at chance level (22.30%). The same results are also confirmed in the simpler scenario with 20 nodes and 3 outbound edges.

Table 8: Undirected graph - shortest path - Comparing results on the second node based on using ground truth or predicted answer of the first node.

| Graph Structure | | | Prediction of First Node | | Prediction of Second Node | | | |
|---|---|---|---|---|---|---|---|---|
| Nodes | Out-degree | Path length | EMN | MEMO | EMN ground truth | EMN predicted answer | MEMO ground truth | EMN predicted annswer |
| 20 | 3 | 3 | 31.99 | 94.40(0.02) | 65.13 | 26.00 | 96.80(0.02) | 93.76(0.08) |
| 20 | 5 | 3 | 23.99 | 69.20(0.07) | 43.00 | 22.30 | 85.38(0.05) | 68.80(0.09) |

Test results for the best 5 hyper-parameters (chosen by training loss) for MEMO, mean and corresponding standard deviation are reported. For EMN we report results from the best run.

## C    BABI

### C.1    TRAINING AND EVALUATION DETAILS

For this experiment we used the English Question Answer dataset Weston et al. (2015). We use the training and test datasets that they provide with the following pre-processing:

- All text is converted to lowercase.

- Periods and interrogation marks were ignored.

- Blank spaces are taken as word separation tokens.

- Commas only appear in answers, and they are *not* ignored. This means that, e.g. for the path finding task, the answer 'n,s' has its own independent label from the answer 'n,w'. This also implies that every input (consisting of 'query' and 'stories') corresponds to a single answer throughout the whole dataset.

- All the questions are stripped out from the text and put separately (given as "queries" to our system).

At training time, we sample a mini-batch of 128 queries from the test dataset, as well as its corresponding stories (which consist of the text prior to the question). As a result, the queries are a matrix of $128 \times 11$ tokens, and sentences are of size $128 \times 320 \times 11$, where 128 is the batch size, 320 is the max number of stories, and 11 is the max sentence size. We pad with zeros every query and group of stories that do not reach the max sentence and stories size.

- For EMN and MEMO, stories and query are used as their naturally corresponding inputs in their architecture.

- In the case of DNC, we embed stories and query in the same way it is done for MEMO. Stories and query are presented in sequence to the model (in that order), followed by blank inputs as pondering steps to provide a final prediction.

- For UT, we embed stories and query in the same way it is done for MEMO. Then, we use the encoder of UT with architecture described in Section H. We use its output as the output of the model.

After that mini-batch is sampled, we perform one optimization step using Adam for all the models that we have run in our experiments, with hyper parameters detailed in Appendix Section D.2. We stop after a fixed number of time-steps, as also detailed in D.2.

Many of the tasks in bAbI require some notion of temporal context, to account for this in MEMO we added a column vector to the memory store with a time encoding derived from Vaswani et al. (2017).

All networks were trained for $2e4$ epochs, each one formed by 100 batch updates.

For evaluation, we sample a batch of $10,000$ elements from the dataset and compute the forward pass in the same fashion as done in training. With that, we compute the mean accuracy over those examples, as well as the accuracy per task for each of the 20 tasks of bAbI. We report average values and standard deviation over the best 5 hyper parameters we used.

## C.2 BABI ABLATIONS

Table 9: bAbI - Ablations

| MEMO Network Architecture | | | | bAbI 10K | |
|---|---|---|---|---|---|
| Positional encoding as in (Vaswani et al., 2017) | Memories keept separated | Recurrent attention | Layernorm | Number of solved tasks | Average error |
| ✓ | ✗ | ✗ | ✗ | 11/20 | 14.81 |
| ✗ | ✓ | ✗ | ✗ | 14/20 | 10.43 |
| ✓ | ✗ | ✓ | ✓ | 17/20 | 4.80 |
| ✗ | ✓ | ✓ | ✗ | 18/20 | 5.00 |
| ✗ | ✓ | ✓ | ✓ | 20/20 | 0.21 |

Results for the best run (chosen by validation set) on the bAbI task. The model was trained and tested jointly on all tasks. All tasks received approximately equal training resources. ✗= not present; ✓= present

## C.3 TASK-WISE RESULTS

Table 10: bAbI Results - average over 5 hyper-parameters with lower loss on the valiation set

| Trial Type | MEMO | MEMO top 5 seeds |
|---|---|---|
| 1 - Single Supporting Fact | 100.00 | 100.00(0.00) |
| 2 - Two Supporting Facts | 100.00 | 99.13(1.78) |
| 3 - Three Supporting Facts | 97.05 | 94.15(6.35) |
| 4 - Two Arg. Relations | 100.00 | 100.00(0.00) |
| 5 - Three Arg. Relations | 100.00 | 100.00(0.00) |
| 6 - Yes/No Questions | 100.00 | 100.00(0.00) |
| 7 - Counting | 100.00 | 96.69(3.57) |
| 8 - Lists/Sets | 100.00 | 99.13(1.94) |
| 9 - Simple Negation | 100.00 | 100.00(0.00) |
| 10 - Indefinite Knowledge | 100.00 | 99.35(1.44) |
| 11 - Basic Coreference | 100.00 | 100.00(0.00) |
| 12 - Conjunction | 100.00 | 100.00(0.00) |
| 13 - Compound Coref | 100.00 | 100.00(0.00) |
| 14 - Time Reasoning | 100.00 | 100.00(0.00) |
| 15 - Basic Deduction | 100.00 | 100.00(0.00) |
| 16 - Basic Induction | 98.75 | 95.05(5.12) |
| 17 - Positional Reasoning | 100.00 | 100.00(0.00) |
| 18 - Size Reasoning | 100.00 | 99.13(1.94) |
| 19 - Path Finding | 100.00 | 100.00(0.00) |
| 20 - Agent's Motivations | 100.00 | 100.00(0.00) |
| Mean error | 0.21 | 0.86(1.11) |
| Solved tasks (>95% accuracy) | 20/20 | n/a |

(Mean and standard deviation of test errors for the best 5 hyperparameters (chosen according the validation loss).

# D   MEMO TRAINING DETAILS AND HYPER-PARAMETERS

## D.1   TRAINING DETAILS

To train MEMO network parameters we use Adam (Kingma & Ba, 2014) with polynomial learning rate decay, starting at $l_start_{memo}$ value, and batch size always equal to 64 for the PAI and shortest path and 128 for bAbI. In all the three tasks MEMO was trained using a cross entropy loss, and the network had to predict the class ID in the paired associative inference task, the node ID in the shortest path problem and the word ID in bAbi.

The halting policy network parameters were updated using RMSProp(Tieleman & Hinton, 2012), with learning rate $l_{halt}$.

The other parameters are reported in Table 11 and 12.

## D.2 FIXED HYPER-PARAMETERS USED ACROSS TASKS

Table 11: Fixed hyper-parameters used across tasks

| Parameter name | Tasks | | | | | | |
|---|---|---|---|---|---|---|---|
| | PAI length 3 | PAI length 4 | PAI length 5 | Shortest path 10-2-2 | Shortest path 20-3-3 | Shortest path 20-5-3 | bAbI |
| I | 32 | 48 | 64 | 20 | 60 | 100 | 320 |
| S | 3 | 3 | 3 | 2 | 2 | 2 | 11 |
| O | 1000 | 1000 | 1000 | 1000 | 1000 | 1000 | 177 |
| $d_c$ | 128 | 128 | 128 | 128 | 128 | 128 | 128 |
| d | 256 | 256 | 256 | 512 | 512 | 512 | 512 |
| $d_a$ | 128 | 128 | 128 | 128 | 256 | 256 | 256 |
| $DropOut_a$ | 0.1 | 0.1 | 0.1 | 0.1 | 0.1 | 0.1 | 0.1 |
| $DropOut_o$ | 0 | 0 | 0 | 0 | 0 | 0 | 0.5 |
| $GRU_R$ hidden size | 256 | 256 | 256 | 256 | 256 | 256 | 256 |
| $MLP_R$ number of layers | 1 | 1 | 1 | 1 | 1 | 1 | 1 |
| $MLP_R$ hidden size | 64 | 64 | 64 | 64 | 64 | 64 | 64 |

## D.3 RANGE OF HYPER-PARAMETERS USED IN SWEEPS

Table 12: Range of hyper-parameters used in sweeps

| Parameter name | Tasks | | | | | | |
|---|---|---|---|---|---|---|---|
| | PAI length 3 | PAI length 4 | PAI length 5 | Shortest path 10-2-2 | Shortest path 20-3-3 | Shortest path 20-5-3 | bAbI |
| $N$ (max number of hops) | [3, 5, 20] | | | [5, 20] | | | [5, 20] |
| $\gamma$ | [0.85, 0.9] | | | [0.9] | | | [0.85, 0.9] |
| $\alpha$ | [1e-4, 1e-2, 0.1] | | | [1e-2, 0.1] | | | [1e-4, 1e-2, 0.1] |
| $\beta$ | [1e-3, 1e-2, 0.1] | | | [1e-2, 0.1] | | | [1e-3, 1e-2, 0.1] |
| $bias_{init}$ | [2, 5] | | | [2, 10] | | | [2, 5] |
| H | 1 | | | [4, 8] | | | [4, 8] |
| $l_s start_{memo}$ | | | | [7e-4, 5e-4, 1e-4] | | | |
| $l_{halt}$ | | | | [1e-4, 5e-5] | | | |

## E    MEMO COMPLEXITY ANALYSIS

In terms of temporal complexity, MEMO has a complexity of $O(n_s \cdot A \cdot N \cdot H \cdot I \cdot S \cdot d)$, where $n_s$ is the number of samples we process with our network, $A$ is the number of answers, $N$ is the upper bound of the number of hops we can take, $H$ is the number of heads used, $I$ is the number of stories, and $S$ is the number of words in each sentence. This is due to the fact that, for each sample, we do the hopping procedure for every answer, taking a number of hops. For each hop we query our memory by interacting with all its slots $I$, for all its size $S \times d$. For all our experiments, all parameters $A$, $N$, $H$, $I$, $S$, $d$ are fixed to constants.

Further, it is worth noting that MEMO is linear with respect to the number of sentences of the input, whereas the Universal Transformer has quadratic complexity.

With respect to spatial complexity, MEMO holds information of all the weights constant, apart from all the context information that needs to be used to answer a particular query. Since the context information is the only one that is input dependent, the spatial complexity is in this case $O(I \cdot S \cdot d)$, which is the size of our memory. In all our experiments, such size is fixed.

## F    ACT DESCRIPTION

We implement ACT as specified in Graves (2016). Based on our implementation of MEMO, we start by defining the *halting unit* $h$ as the following:

$$h_t = \sigma(\pi_t) \tag{19}$$

where $\pi_t$ is the binary policy of MEMO. This is slightly different than the original ACT which represents such unit with:

$$h_t = \sigma(W_h s_t + b_h) \tag{20}$$

where $W_h$ and $b_h$ are trainable weights and biases respectively, and $s_t$ is the previous observed state. We argue that this slight change increases the fairness of the comparison, for two reasons: firstly, $\pi_t(a|s_t, \theta)$ depends on $s_t$, but it uses several non-linearities to do so, rather than it being a simple linear projection, so it should enable more powerful representations. Secondly, this makes it much more similar to our model while still being able to evaluate the feasibility of this halting mechanism.

From this point we proceed as in the original work by defining the *halting probability*:

$$p_t = \begin{cases} R & \text{if } t = T \\ h_t & \text{otherwise} \end{cases} \tag{21}$$

where

$$T = min\{t' : \sum_{t=1}^{t'} h_t >= 1 - \epsilon\} \tag{22}$$

where $\epsilon$, as in Graves (2016) is fixed to be $0.01$ in all experiments. The reminder $R$ is defined as:

$$R = 1 - \sum_{t=1}^{T-1} h_t \tag{23}$$

Finally, the answer provided by MEMO+ACT is defined as:

$$a = \sum_{t=1}^{T} p_t a_t \tag{24}$$

where $a_t$ corresponds to the answer that MEMO has provided at hop $t$.

## G    DNC ARCHITECTURE AND HYPERPARAMETERS

We use the same architecture as described in Graves et al. (2016), with exact sizes of each layer described on Table 13.

We also performed a search on hyperparameters to find such hyperparameters, with ranges reported on Table 14.

| Parameter name | Value |
|---|---|
| Optimizer algorithm | Adam |
| Learning rate | 0.0005 |
| Input embedding size | 128 |
| Input pondering steps | 10 |
| Controller type | LSTM |
| Controller hidden size | 128 |
| Memory number of read heads | 3 |
| Memory word size | 384 |
| Memory output size | 128 |

Table 13: Hyperparameters used on all tasks trained with DNC.

| Parameter name | Value |
|---|---|
| Learning rate | {0.0001, 0.0005} |
| Input pondering steps | {0, 5, 10} |
| Memory number of read heads | {1, 3} |

Table 14: Hyperparameters ranges used to search over with DNC.

## H  UNIVERSAL TRANSFORMER ARCHITECTURE AND HYPERPARAMETERS

We use the same architecture as described in Dehghani et al. (2018). More concretely, we use the implementation and hyperparameters described as 'universal_transformer_small' that is available at `https://github.com/tensorflow/tensor2tensor/blob/master/tensor2tensor/models/research/universal_transformer.py`. For completeness, we describe the hyperpameters used on Table 15.

We also performed a search on hyperparameters to train on our tasks, with ranges reported on Table 16.

| Parameter name | Value |
|---|---|
| Optimizer algorithm | Adam |
| Learning rate | 5e-4 |
| Input embedding size | 128 |
| Attention type | Dot product |
| Attention hidden size | 512 |
| Attention number of heads | 8 |
| Transition function | Fully connected neural network |
| Transition hidden size | 128 |
| Number of hidden layers | 6 |
| Recurrence type | 'RNN' (not LSTM) |
| Number of recurrent steps | 6 |
| Attention dropout rate | 0.1 |
| ReLU dropout rate | 0.1 |
| Layer pre/postprocess dropout | 0.1 |

Table 15: Hyperparameters used for all experiments for UT.

| Parameter name | Value |
|---|---|
| Learning rate | {1e-3, 5e-4, 1e-4} |
| Number of hidden layers | {2, 6} |
| Attention hidden size | {128, 512} |
| Transition hidden size | {512, 128} |
| Attention number of heads | {4, 8} |

Table 16: Hyperparameters ranges used to search over with UT.

