# OpenReview forum: "MEMO: A Deep Network for Flexible Combination of Episodic Memories"
_ICLR.cc/2020/Conference — Accept (Poster)_

### Official Review · AnonReviewer2 · 2019-10-29
**Official Blind Review #2**

**Rating:** 8

**Review:**

Summary:

This paper proposes two main changes to the End2End Memory Network (EMN) architecture: a separation between facts and the items that comprise these facts in the external memory, policy to learn the number of memory-hops to reason. The paper also introduces a new Paired Associative Inference (PAI) task inspired by neuroscience and shows that most of the existing models including transformers struggle to solve this task while the proposed architecture (called MEMO) solves it better. MEMO also works well in the shortest path finding tasks and bAbI tasks.

My comments:

Overall, I see this paper as an improvement over EMN. The proposed PAI task can be seen as an example task where Transformers struggle while recurrent architectures learn better. Interestingly, the authors use a separate halting policy network to reduce computation time.

1. Section 2.1 requires more clarity. There is a confusion in the usage of I and S. I represents the number of stories or the number of sentences in the stories?
2. Scaling up NTM/DNC to larger memory work was done in Sparse Access Memory (SAM) by Rae et al. 2016. This needs to be included in section 3.1
3. In Table 2, why is the prediction of the second node easier for the model than the prediction of the first node? I see this trend only for EMN, UT, DNC. Not in MEMO.
4. Are the authors willing to release the code and data to reproduce their results?

Minor comments:

1. Page 2, second para: ENM should be EMN.
2. Vec inverse in Eqn 14 was never introduced.
3. Table 3: The notation of (20/20) was never introduced. I can guess what it means. But please be explicit.

===============
After rebuttal: Authors have addressed all my questions. I  recommend  "Accept".


**Experience Assessment:**

I have published one or two papers in this area.

**Review Assessment: Checking Correctness Of Derivations And Theory:**

N/A

**Review Assessment: Checking Correctness Of Experiments:**

I assessed the sensibility of the experiments.

**Review Assessment: Thoroughness In Paper Reading:**

I read the paper thoroughly.

---

> ### Author Response · Authors · 2019-11-08
> **Answer to reviewer #2**
>
> We thank the reviewer for their positive comments. Below we will try to address the comments one by one:
>
> 1) We agree with the reviewer that this was confusing. To be clear, for instance in bAbI, I is the number of stories and S is the number of words in each sentence. We changed the text in the paper to clarify this. We also clarified the vector notation and corrected a few other typos we found in section 2. This is all in the updated version of the paper.
>
> 2)We mentioned sparse DNC, but erroneously omitted the  reference. This is now corrected in the updated version of the paper, thanks for spotting this.
>
> 3) Regarding Table 3, we thank the reviewer for pointing this out, we believe this will greatly contribute to clarifying the discussion of the results for the shortest path task. The answer to the question is based on how the models were trained. As highlighted in Appendix B.1, due to the expectations that each method has with respect to its input, we use the following training regimes:
> For Universal Transformer (UT), the model was trained to produce both node answers based on receiving as input a concatenation of the graph description and the query.
> DNC receives both the graph description and the query as a sequence, and proceeds, after some pondering, to produce the sequence of answers for both nodes in the path.
> EMN and MEMO embed the graph description in their memory, and then they use the query to produce the answer for the first node, then only after this the model produces the answer to the second node. However, the paper was not clear about a critical difference between EMN and MEMO. To predict the second node we used a different strategy for the models. For EMN we use the ground truth answer of the first node as the query for the second node, whereas for MEMO we used the answer predicted by the model for the first node as the query for the second node.
> Given this training regime: DNC and UT have a ‘global view’ on the problem in order to provide an answer for the second node. This means that, to answer the second node in the path, they can still reason and work backwards from the end node, and while still having information about the initial node in the path. This makes it intuitive for them to achieve better performance in the second node, as it is closest to the end node of the path, so less reasoning is needed to achieve good performance. On the contrary, MEMO has a ‘local view’ on the problem, the answer to the second node depends on the answer about the first node. Therefore, it cannot do better than chance if the answer to the first node is not correct.
> Also, in the table presented in the paper, EMN increases the performance on the second node because we use the ground truth answer of the first node as the query for the second node, and because the answer to the second node it is easier given that it is closer to the end node, EMN increases its performance compared to the first node.
>
> This training regime was used to enhance the performance of the baselines as much as possible, whereas for MEMO we kept to a principled approach to test its capabilities of to sequentially reasoning over multiple steps problems.
>
> To better answer the reviewer, we also ran another experiment where we tested MEMO and EMN in both conditions:
> The ground truth answer of the first node was used as the query for the second node.
> The answer predicted by the model for the first node was used as the query for the second node.
> The results are now summarised in Table 8 in Appendix B.2 and in the case of 20 Nodes with 5 outbound edges, we can see that if we give MEMO the ground truth for node 1 as query for node 2 the performance increases from the one related to the prediction of the first node (85.38%(0.05) vs. 69.20%(0.07)). Interestingly, if we use for EMN the same training regime used for MEMO - i.e. the prediction is used to query the second node - then EMN perform almost at chance level (22.30%). The same results are also confirmed in the simpler scenario with 20 nodes and 3 outbound edges (see Table 8, Appendix B.2).
> We hope that with this explanation and the additional experiment we performed are helping to clarifying the reviewers’ doubts.
>
> 4)We are going to release the new datasets for the PAI task before the ICLR conference. Regarding the code, given that the model is based on EMN and it is explained in greater details in the paper we think it would be easy to implement.
>
> 5) We have corrected all the minor comments that the reviewer pointed out.

---

### Official Review · AnonReviewer1 · 2019-10-30
**Official Blind Review #1**

**Rating:** 6

**Review:**

Summary:

This paper presents a new task (paired associate inference), drawn from cognitive psychology, which requires linking many pieces of information together to make inferences with long range dependencies. Experimental results show that standard memory architectures fail on these tasks. To redress this, the paper proposes a new memory architecture with several new features that allow for much better performance on the paired associate task. Finally, the paper undertakes systematic experiments on more traditional domains like shortest path problems, showing that the new architecture achieves modest improvements.

Major comments:

Overall this is an interesting and useful work which uses a task from cognitive psychology to illuminate reasoning limitations in prior memory augmented neural networks. It then proposes several architectural and algorithmic fixes to improve performance. The resulting solution appears to work based on the experimental evaluation, and although this does not lead to substantial improvements in the state of the art on the bAbI dataset, it seems likely that these improvements could be useful for even harder task settings.

The new algorithmic improvements are justified mainly by their performance in experiments, which is completely acceptable if the experiments have been done to a high standard. Nevertheless, it would be wonderful if more insight could be gained about why exactly this configuration of architectural changes was selected and what they contribute. The ablation experiments go some way to addressing this question and perhaps the paper would benefit from a greater discussion of these results in the main text.

The experiments appear to have been done to a high standard, and it is promising that the method’s advantages are more pronounced as the task gets harder (eg by demanding longer chains of inference).

The paper could also be improved by more clearly describing how hyper parameters were selected for each experiment (including the different ablation studies). Why were hyperparamters sometimes chosen based on training loss and other times chosen based on validation loss? In general it would be useful to explain the selection procedure very carefully for each result, since it appears to have differed between evaluations.

The paper is reasonably easy to follow, but could be streamlined to enable more discussion of material that is currently in the appendix.

Typos:

ENM->EMN on pg2


**Experience Assessment:**

I have read many papers in this area.

**Review Assessment: Checking Correctness Of Derivations And Theory:**

I assessed the sensibility of the derivations and theory.

**Review Assessment: Checking Correctness Of Experiments:**

I assessed the sensibility of the experiments.

**Review Assessment: Thoroughness In Paper Reading:**

I read the paper at least twice and used my best judgement in assessing the paper.

---

> ### Author Response · Authors · 2019-11-08
> **Answer to reviewer #1**
>
> We would like to thank the reviewer for their thoughtful comments. Regarding the comments we will try to address them step by step:
>
> With respect to the motivation for each novel architectural component, we will address each in turn. The first novel component is the separation of all memory elements within the input. This contrasts with previous SOTA methods which combine word embedding with a positional encoding scheme (Sukhbaatar et al., 2015; Santoro et al., 2017; Vaswani et al., 2017; Pavez et al., 2018; Devlin et al., 2018).  This was done because the focus of MEMO is on the ability to form long-range inference across distinct episodes for which it is required the ability to find relations amongst elements shared across temporally distinct episodes. The crucial insight here is that inferential reasoning depends on representing the individual episode elements, and using these to discover relations across the episodes by doing a sequence of reasoning hops and stop when enough evidence has been accumulated (our second component). The idea is that by passing the full set of memory elements as an input to the powerful recursive attention mechanism (hops) the model has the power to weight each single element differently and then recombine these to answer previously unseen trials. In this respect, our ablation experiments confirm that it is the combination of the memory representations in combination with the  recursive attention that is needed to perform inference trials (see Table 7, row 4). Interestingly, each of these two components alone was not sufficient (see Table 7, row 2 and 3). Moreover, this conclusion was only valid for inference trials, whereas direct trials, which are tests of pure episodic memories (i.e. no inference is required) could be solved even without these changes (see Figure 3, c&d). These ablations results, also confirmed for bAbI (see Appendix C.2), validate our motivations.
> Also, as pointed out in the second paragraph of the introduction, our motivations have roots in neuroscientific findings (Kumaran & McClelland, 2012; Banino et al., 2016; Schapiro et al., 2017; Koster et al., 2018). Moreover, as pointed out in Figure 2 and section 5.2, our model seems to replicate the retrieval mechanism predicted by computational models of the hippocampus (Kumaran & McClelland, 2012) and later confirmed by empirical experiments on human subjects (Koster et al., 2018).
> We hope to have clarified our motivations for having chosen this particular architecture, however if that is not the case we are happy to accept suggestions from the reviewer on how we can make this more clear in the paper.
>
> We thank the reviewer for thinking that our experimental procedures meet high standards. Regarding the choice of hyperparameters, we agree that the paper was confusing. For the Paired Associative Inference and bAbI selection was done on the validation set. For the shortest path problem we used the training loss. This choice of the training loss was done because the path problems are generated procedurally. The same procedure was applied to all the models. This has now been clarified in the paper.
>
> We agree with the reviewer that some parts of the text could be moved from the appendix in the main text. However the ICLR conference committee promoted rules to stick to page limit, and so we had to make some decision to move some results in the appendix. With this in mind we believe we could stick to such rules while still being able to elaborate on the contributions. So now we made some clarification in section 2 to help the reader understand the model, and now all the Tables in the appendix are hyperlinked in the main text to make it easier to read them.
>
> Thanks for spotting the typos, these have now been fixed.

---

### Decision · Program_Chairs · 2019-12-19

**Decision:**

Accept (Poster)

**Comment:**

The authors introduce a new associative inference task from cognitive psychology, show shortcomings of current memory-augmented architectures, and introduce a new memory architecture that performs better with respect to the task. The reviewers like the motivation and thought the experimental results were strong, although they also initially had several questions and pointed to areas of the paper which lacked clarity. The authors updated the paper in response to the reviewer's questions and increased the clarity of the paper. The reviewers are satisfied and believe the paper should be accepted.